# POLICY OPTIMIZATION BY LOCAL IMPROVEMENT THROUGH SEARCH

## ABSTRACT

Imitation learning has emerged as a powerful strategy for learning initial policies that can be refined with reinforcement learning techniques. Most strategies in imitation learning, however, rely on per-step supervision either from expert demonstrations, referred to as behavioral cloning (Pomerleau, 1989; 1991) or from interactive expert policy queries such as DAgger (Ross et al., 2011). These strategies differ on the state distribution at which the expert actions are collected – the former using the state distribution of the expert, the latter using the state distribution of the policy being trained. However, the learning signal in both cases arises from the expert actions. On the other end of the spectrum, approaches rooted in Policy Iteration, such as Dual Policy Iteration (Sun et al., 2018b) do not choose next step actions based on an expert, but instead use planning or search over the policy to choose an action distribution to train towards. However, this can be computationally expensive, and can also end up training the policy on a state distribution that is far from the current policy's induced distribution. In this paper, we propose an algorithm that finds a middle ground by using Monte Carlo Tree Search (MCTS) (Kocsis & Szepesvári, 2006) to perform local trajectory improvement over rollouts from the policy. We provide theoretical justification for both the proposed local trajectory search algorithm and for our use of MCTS as a local policy improvement operator. We also show empirically that our method (Policy Optimization by Local Improvement through Search or POLISh) is much faster than methods that plan globally, speeding up training by a factor of up to 14 in wall clock time. Furthermore, the resulting policy outperforms strong baselines in both reinforcement learning and imitation learning.

## 1 INTRODUCTION

Reinforcement learning (RL) has seen a great deal of success in recent years, from playing games (Mnih et al., 2015; Silver et al., 2016) to robotic control (Gu et al., 2017; Singh et al., 2019). These successes showcase the power of learning from direct interactions with environments. However, a well-known disadvantage of reinforcement learning approaches is the demand for a large number of samples for learning; for example, OpenAI Five learned to solve DOTA using Proximal Policy Optimization (PPO) (Schulman et al., 2017) by playing 180 years worth of games against itself daily (OpenAI, 2018). The issue with sample complexity can be mitigated by learning from expert behavior. For example, AlphaStar (The Alpha Star Team, 2019) uses models that are pre-trained using supervised learning techniques on expert human demonstrations, before RL is used for refinement.

A variety of strategies have been developed in imitation learning to learn from expert behavior, where the expert can be a human (Stadie et al., 2017) or a pre-trained policy (Ho & Ermon, 2016). Earlier approaches, such as behavioral cloning (BC), relied on training models to mimic expert behavior at various states in the demonstration data (Pomerleau, 1991). However, these models suffer from the problem that even small differences between the learned policy and the expert behavior can lead to a snow-balling effect, where the state distribution diverges to a place where the behavior of the policy is now meaningless since it was not trained in that part of space (Ross & Bagnell, 2010). In order to mitigate these issues, DAgger (Ross et al., 2011) uses an expert policy to provide supervision to the policy being learned. While these strategies differ on the state distribution at which the expert actions are optimized – for example BC uses the state distribution of the expert, DAgger-like approaches

use the state distribution from the policy being trained – the supervision signal itself is the expert's behavior at each step.

On the other end of the spectrum, approaches rooted in policy iteration, such as Dual Policy Iteration (Sun et al., 2018b) do not mimic next step actions of a policy directly, but instead use planning or search over the policy to choose an action distribution to train towards (Silver et al., 2017). However, this can be computationally expensive, and can also end up training the policy on a state distribution that is far from the current policy's induced distribution.

In this paper, we propose an algorithm that finds a middle ground by using Monte Carlo Tree Search (MCTS) (Kocsis & Szepesvári, 2006) to perform local trajectory improvement over states sampled from different time steps in the trajectory unrolled from the policy. This approach has the benefit of training the policy on a state distribution that is close to that induced by the policy itself, while using local search or planning over a smaller horizon to generate a good policy to train towards. This approach stands in contrast to other works in interactive imitation learning that correct distribution mismatch by using one-step feedback. We provide theoretical justification for the advantage of a balanced local trajectory improvement and show that MCTS can serve as a policy improvement operator under certain conditions[1].

An added benefit of our effort is computational – depth parallel MCTS on local trajectory segments is much faster than traditional sequential MCTS to generate demonstrations. We show that our proposed algorithm can easily be parallelized to enable more efficient imitation learning from MCTS. Notably, this level of parallelism is present at varying depths which differs from existing works on parallel MCTS (Chaslot et al., 2008).

In summary, our main contributions are:

- A general interactive imitation learning algorithm that balances the expert feedback quality and the state distribution divergence. Our proposed approach provides a flexible local trajectory improvement based on MCTS.
- Theoretical analysis on the general benefit of local improvement and specific case study on using MCTS as a local improvement operator.
- Strong empirical performance on a suite of high-dimensional continuous control problems based on both sample efficiency and training time.

## 2 RELATED WORK

**Imitation Learning.** Imitation learning (IL) refers to the problem of learning to perform a task from expert demonstrations. Behavioral cloning (Widrow & Smith, 1964) is one popular approach which maximizes the likelihood of expert actions under the agent policy (Pomerleau, 1989; Schaal, 1999; Muller et al., 2006; Mlling et al., 2013; Bojarski et al., 2016; Giusti et al., 2016; Mahler & Goldberg, 2017; Wang et al., 2019; Bansal et al., 2018). Inverse Reinforcement Learning is another popular form of IL where a reward function is extracted from expert demonstrations and then a policy is trained to maximize that reward (Ziebart et al., 2008; Finn et al., 2016; Fu et al., 2018; Ho & Ermon, 2016).

In this work we focus on imitation learning via behavioral cloning. Despite its success in canonical problems, such as Go (Silver et al., 2017) and Starcraft (The Alpha Star Team, 2019), behavioral cloning suffers from many challenges, most notably distributional shift (Daumé et al., 2009).

To explain distributional shift, let us assume we train an agent by performing supervised learning on the actions an expert has taken. A small error in the first time step of training may bring the agent to a state that the expert has rarely visited and is therefore not as well modeled. Over time, this error compounds, leading the agent to states far from expert behavior and diverging further from the expert demonstrations. The longer the episode is, the more likely the agent is to deviate from expert demonstrations.

---

[1] We note that other techniques such as Generative Adversarial Imitation Learning (GAIL) have been developed that use example demonstrations in a modified objective which is still trained by interacting with the environment (Ho & Ermon, 2016). These approaches directly operate on the induced state distribution itself; we do not consider these approaches here

Various solutions to the problem of distributional shift have been proposed (Daumé et al., 2009; Ross & Bagnell, 2010; Ross et al., 2011; Ho & Ermon, 2016; Laskey et al., 2017; Bansal et al., 2018; de Haan et al., 2019). Some of these approaches reduce the effects of distributional shift by iteratively querying the expert (Daumé et al., 2009; Ross et al., 2011). DAgger (Ross et al., 2011), one of the most widely used of these solutions, queries the expert on each of the states that are visited by the policy and uses expert demonstrations to improve the policy.

At the other extreme, we have Policy Iteration approaches, such as Dual Policy Iteration (Sun et al., 2018b), AlphaZero (Silver et al., 2017) and ExIT (Anthony et al., 2017) that do not directly mimic the actions of an expert, but instead plan or search over the policy to choose an action distribution to train towards (Silver et al., 2017). These methods can be computationally expensive and their long planning horizon can cause the state distribution to diverge far from the current policy's induced distribution.

In this paper, we propose a new algorithm, named POLISh, that strikes a middle ground by performing multi-step improvements over states sampled from different time steps in the trajectories generated by the policy. We use MCTS (Kocsis & Szepesvári, 2006) to perform the local trajectory improvements. We show the promise of our approach with both theoretical and empirical results.

**Monte Carlo Tree Search.** We refer the reader to (Browne et al., 2012) for a comprehensive survey on MCTS. To generate the proposed locally optimized trajectories, we follow recent work that explores using MCTS to provide feedback for policy improvement (Guo et al., 2014; Silver et al., 2017; Anthony et al., 2017). MCTS is a best-first tree search method which uses a policy to explore the most promising actions first. Through repeated simulation, MCTS builds a tree whose nodes represent states and whose branches correspond to actions that can be taken from those states. The objective is to maximize total return, so after all simulations have been completed, a final trajectory is generated by traversing the most visited sequence of nodes.

## 3  BACKGROUND & PRELIMINARIES

**Markov Decision Processes.**  We consider policy learning for Markov Decision Processes (MDPs) represented as a tuple $(\mathcal{S}, \mathcal{A}, \mathcal{P}, r, \gamma, \mathcal{D})$. Let $\mathcal{S}$ denote the state space, $\mathcal{A}$ the action space, $\mathcal{P}(s'|s, a)$ the (probabilistic) state dynamics, $r(s, a)$ the reward function, $\gamma$ the discount factor and $\mathcal{D}$ the initial state distribution.

**Policy Learning.**  A stochastic policy $\pi$ maps a state $s \in \mathcal{S}$ to a distribution over the actions $\mathcal{A}$, denoted by $\pi(s)$. At each state, an action $a$ is sampled from $\pi(s)$ with probability $\pi(a|s)$ and a reward of $r(s, a)$ is received by $\pi$. The goal is to learn a policy that maximizes the accumulated discounted rewards $J_{\mathcal{D}}(\pi) = \mathbb{E}_{\tau \sim \pi}[\sum_{i=0}^{\infty} \gamma^i r(s_i, a_i)]$. We omit the dependency of the initial state distribution $\mathcal{D}$ when the context is clear. A few useful quantities related to a policy are the value function $V_\pi$, the state-action value function $Q_\pi$ and the advantage function $A_\pi$ defined as follows:

$$V_\pi(s) = \mathbb{E}_{\tau \sim \pi}[\sum_{i=0}^{\infty} \gamma^i r(s_i, a_i)|s_0 = s],$$

$$Q_\pi(s, a) = \mathbb{E}_{\tau \sim \pi}[\sum_{i=0}^{\infty} \gamma^i r(s_i, a_i)|s_0 = s, a_0 = a],$$

$$A_\pi(s, a) = Q_\pi(s, a) - V_\pi(s)$$

As the policy takes a sequence of actions, its performance has a strong connection to the state distribution induced by its actions. We define quantities related to the state distributions induced by policies at different time steps. Let $d_\pi^t$ denote the state distribution obtained by following $\pi$ for $t$ steps. We use $d_\pi = (1-\gamma) \sum_{t=0}^{\infty} \gamma^t d_\pi^t$ to denote the accumulated discounted state distribution. With these quantities, we can rewrite $J(\pi) = \sum_{t=0}^{\infty} \mathbb{E}_{s \sim d_\pi^t, a \sim \pi(s)}[\gamma^t r(s, a)] = \mathbb{E}_{s \sim d_\pi, a \sim \pi(s)}[r(s, a)]$.

We will make use of the following relationship between the two policies in our analysis (Kakade & Langford, 2002):

$$J(\pi') = J(\pi) + \mathbb{E}_{s \sim d_\pi}[\mathbb{E}_{a \sim \pi'(s)}[A_\pi(s, a)]] \tag{1}$$

Throughout the paper, we use $\pi^*$ to denote the expert policy.

# 4   THE POLISH ALGORITHM

We first present a general framework for imitation learning from local trajectory improvements in Section 4.1. We then describe a method based on MCTS to perform the proposed local trajectory optimization in Section 4.2.

## 4.1   MAIN ALGORITHM

Our main algorithm builds on the observation that behavioral cloning (Pomerleau, 1991) and DAgger (Ross et al., 2011) are two extremes along a spectrum of possible algorithms using experts to generate trajectory improvement. This is illustrated in Figure 1. The solid nodes represent the states visited by a current policy $\pi$ during a rollout of length $T$. DAgger seeks to collect expert feedback on every state along $\tau$, while behavioral cloning only imitates a single complete trajectory from the expert. DAgger collects 1-step demonstrations from every state while behavioral cloning collects $T$-step from a single step. It is easy to see that there is a spectrum of algorithms that lie in between, parametrized by the time horizon $t$ to collect expert demonstrations and the frequency with which to choose the start state for the expert policy.

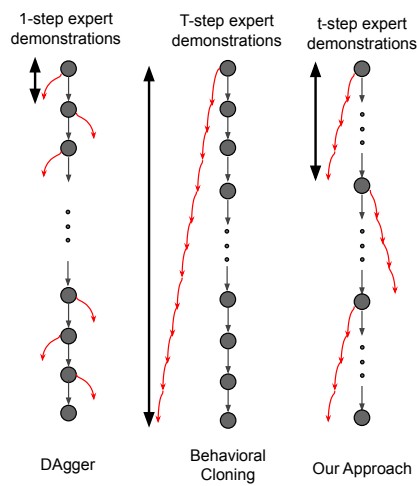

Figure 1: DAgger (Ross et al., 2011) collects 1-step feedback at every state. Behavioral cloning collects $T$-step feedback only from the initial state. Our method collects $t$-step feedback at the frequency of every $t$ states.

We present our main algorithm in Algorithm 1 that collects locally improved partial trajectories, as shown in Figure 1 and trains a policy to imitate the expert on the collected trajectories. For each iteration, we first collect trajectories from a current policy $\pi_i$ (Line 3). Then, we collect every $k$ states from each trajectory to use as starting states for running the expert policy (Line 7, 8, 9). From each state collected, we rollout $\pi^*$ for $t$ steps to generate a partial trajectory (Line 12, 13). Once all the data has been generated, we optimize for the behavioral cloning loss $L(D, \pi) = \frac{1}{|D|} \sum_{(s,a^*) \in D} (I(\pi(s) \neq a^*))$, i.e., the supervised learning loss from data collected in $D$, to generate a new policy and repeat the process.

## 4.2   MCTS AS THE EXPERT POLICY

General imitation learning requires having access to an expert policy. We follow a series of recent works that explore applying MCTS to provide feedback for policy improvement (Guo et al., 2014; Silver et al., 2017; Anthony et al., 2017). Specifically, for every iteration, $\pi^*$ becomes the policy obtained by running MCTS with $\pi_i$ by following a variant of MCTS that uses a slightly modified UCT rule to select leaf node in MCTS: $\mathrm{UCT}(s, a) = \frac{r(s,a)}{n(s,a)} + c \cdot \pi(a|s) \sqrt{\frac{\log n(s)}{n(s,a)}}$, where $n(s)$ is the number of times the node $s$ has been visited so far, $n(s, a)$ is the number of times action $a$ is selected at node $s$, and $r(s, a)$ is the sum of all rewards obtained in simulations by taking $a$ in $s$. At a leaf node, Monte Carlo rollouts for value estimation (Browne et al., 2012) is replaced by an estimate obtained from a value network trained with the policy. This approach removes the need for a pre-defined expert policy and can be effective if MCTS can serve to improve the policy. We provide a sufficient condition that justifies this design choice.

# 5   THEORETICAL ANALYSIS

In this section, we provide theoretical justification for considering local trajectory improvement in relation to both behavioral cloning and DAgger. We then prove that, under certain conditions, MCTS

---

**Algorithm 1** POLISH (Policy Optimization by Local Improvement through Search)

---

1: **Input:** an initial policy $\pi_1$, the expert policy $\pi^*$, the segment length $t$, number of iterations $N$, a policy class $\Pi$.
2: **for** $1 \leq i \leq N$ **do**
3:    Sample $\{\tau_j\}_{j=1}^n$ from $\pi_i$
4:    $S \leftarrow \emptyset$
5:    $D \leftarrow \emptyset$
6:    **for** each trajectory $\tau_j = (s_0, a_0, s_1, \cdots, s_{T-1}, a_{T-1}, s_T)$ **do**
7:      $k \leftarrow \lceil T/t \rceil - 1$
8:      $S_j \leftarrow \{s_0, s_t, s_{2t}, \cdots, s_{kt}\}$
9:      $S \leftarrow S \cup S_j$
10:    **end for**
11:    **for** each $s \in S$ **do**
12:      Run $\pi^*$ for $t$ steps from the state $s$ to generate a partial trajectory $\tau_s = (s, a'_0, s'_1, \cdots, a'_{t-1}, s'_t)$
13:      $D \leftarrow D \cup \{(s, a'_0), (s'_1, a'_1), \cdots, (s'_{t-1}, a'_{t-1})\}$
14:    **end for**
15:    $\pi_{i+1} \leftarrow \arg\min_{\pi \in \Pi} L(D, \pi)$
16: **end for**
17: **return** $\pi_{N+1}$

---

can serve as a method for improving local trajectories, thus providing a theoretical foundation for previous empirical works (Silver et al., 2017; Anthony et al., 2017). We provide our proofs in the Appendix.

### 5.1 MOTIVATION FOR LOCAL TRAJECTORY IMPROVEMENT

Imitation learning only makes sense if the policy we are imitating is better than our current policy. We formalize below what "better" means in this context:

**Assumption 1.** *Assume that $\pi^*$ is an expert policy such that for every state $s \in \mathcal{S}, \mathbb{E}_{a \sim \pi^*(s)}[Q_\pi(s,a)] \geq \mathbb{E}_{a \sim \pi(s)}[Q_\pi(s,a)]$ and $\mathbb{E}_{a \sim \pi^*(s), s' \sim \mathcal{P}(\cdot|s,a)}[V_{\pi^*}^k(s')] \geq \mathbb{E}_{a \sim \pi(s), s' \sim \mathcal{P}(\cdot|s,a)}[V_{\pi^*}^k(s')]$ for every $k \geq 1$, where $V_\pi^k$ denotes the accumulated discounted $k$-step returns.*

The first assumption ensures that $J(\pi^*) \geq J(\pi)$ according to Equation 1. The second assumption ensures that one-step deviation from $\pi^*$ according to $\pi$ will result in lower-valued states if we were to follow $\pi^*$ from that point on. This is typically true if $\pi$ is a greedy policy so it pays less attention to future rewards.

To motivate the idea of using local trajectory improvement instead of complete trajectory imitation learning, we extend the analysis in (Ross & Bagnell, 2010) on behavioral cloning. A major issue with (long-horizon) behavioral cloning is the potential for cascading errors if a learned policy makes a mistake early on. This compounding behavior is captured in the following theorem:

**Theorem 1.** *(Theorem 2.1 in (Ross & Bagnell, 2010), rewritten in terms of rewards instead of costs) Let $\pi$ be a policy such that $\epsilon = \mathbb{E}_{s \sim d_{\pi^*}}[e_\pi(s)]$. Then $J(\pi) \geq J(\pi^*) - T^2 \epsilon$, where $T$ is the task horizon and $e_\pi(s) = \mathbb{E}_{a \sim \pi(s), a* \sim \pi^*(s)}[I(a \neq a^*)]$.*

The cascading error effect manifests as a quadratic function in the task horizon. We can do a more careful analysis that motivates the design of an algorithm that clones behaviors at a shorter time scale. Define $J_\pi^{t_i:t_{i+1}}(\pi')$ to be the rewards obtained by following $\pi'$ from time step $t_i$ to $t_{i+1}$ from the start distribution $d_\pi^{t_i-1}$. Assume we divide the time horizon into equal parts with length $t$ so that we have $T/t$ segments with $t_i = it$.

$$J(\pi) = \sum_{i=1}^{T/t} \gamma^{t(i-1)} J_\pi^{t_i:t_{i+1}}(\pi) \geq \sum_{i=1}^{T/t} \gamma^{t(i-1)} \left( J_\pi^{t_i:t_{i+1}}(\pi^*) - t^2 \epsilon_i \right) \tag{2}$$

where $\epsilon_i$ is the error rate computed between steps $t_i$ and $t_{i+1}$. Assuming $\epsilon_i = \epsilon$ holds across different segments, the expression above simplifies to

$$J(\pi) \geq \sum_{i=1}^{T/t} \gamma^{t(i-1)} J_\pi^{t_i:t_{i+1}}(\pi^*) - \sum_{i=1}^{T/t} \gamma^{t(i-1)} t^2 \epsilon = \sum_{i=1}^{T/t} \gamma^{t(i-1)} J_\pi^{t_i:t_{i+1}}(\pi^*) - \frac{1-\gamma^T}{1-\gamma^t} t^2 \epsilon \quad (3)$$

The objective is to maximize $J(\pi)$ and this is done by optimizing the lower bound on the right hand side via imitating $\pi^*$. Next, we consider how the two terms on the right hand side interact.

**Proposition 1.** *Under Assumption 1, $\sum_{i=1}^{T/t} \gamma^{t(i-1)} J_\pi^{t_i:t_{i+1}}(\pi^*) \leq \sum_{i=1}^{T/t'} \gamma^{t'(i-1)} J_\pi^{t'_i:t'_{i+1}}(\pi^*)$ if $t'$ is a multiple of $t$. Furthermore, it is maximized at $t = T$.*

This proposition says that by multiplying the length of a segment, the first term grows.

**Proposition 2.** $\frac{1-\gamma^T}{1-\gamma^t} t^2 \epsilon$ *is an increasing function of $t > 0$ for $\gamma \in (0, 1)$.*

So the second term is also an increasing function in $t$, there can be a balance point where the right hand side is maximized, giving the tightest lower bound on the performance of policy $\pi$.

There is empirical evidence, (Silver et al., 2017; Anthony et al., 2017), that fitting to the expert action distribution is a more robust approach than fitting to a single action that the expert takes. In Proposition 2, we show that we can relate $\epsilon$ to the KL-divergence between the action distributions of the policy and the expert.

**Proposition 3.** $\epsilon = \mathbb{E}_{s \sim d_{\pi^*}}[e_\pi(s)] \leq \mathbb{E}_{s \sim d_{\pi^*}}[D_{KL}(\pi(s)||\pi^*(s)) + H(\pi(s))]$, *where $D_{KL}$ is the KL-divergence between two distributions and $H$ is the entropy of a distribution.*

As a result, by optimizing the objective presented in Section 4.1, we are improving the lower bound presented in Equation 3.

## 5.2 MONTE CARLO TREE SEARCH AS A LOCAL TRAJECTORY IMPROVEMENT OPERATOR

Now we turn our attention to $\pi^*$. In traditional imitation learning, the expert policy $\pi^*$ can be a human expert (Ross et al., 2011; Stadie et al., 2017) or a pre-trained policy (Ho & Ermon, 2016). In either case, $\pi^*$ is considered fixed and independent of the current learned policy. Recently, several works (Guo et al., 2014; Silver et al., 2016; Anthony et al., 2017) explored using MCTS to plan ahead when a *generative model* of the underlying MDP is available (Kearns et al., 2002; Kakade et al., 2003) and thus produce better action decisions compared with directly following a policy. This approach circumvents the requirement of having a pre-defined expert and has the nice property of bootstrapping from the existing policy (Sun et al., 2018b). However, it is not clear whether MCTS can always produce better trajectories for demonstrations. Another complication is the modified MCTS procedure, whose changes include that the UCT algorithm weights the exploration term by the policy prior probability and actual Monte Carlo rollout estimation of a state value is replaced by a value network.

Let $V_\pi^e$ be a value network that estimates the true value function $V_\pi$ for $\pi$. During MCTS, whenever a value estimation for a leaf node is needed, the value network is used. We can derive a $Q$-function estimation $Q_\pi^e$ for $\pi$ by using the following equation $Q_\pi^e(s, a) = \mathbb{E}[r(s, a) + \gamma V_\pi^e(s')]$; that is, we use the sampled rewards and value estimation for the next state to form an estimation for the Q-function, where the expectation is taken over the stochasticity of the environment transitions. The following theorem, for which we provide a proof in the appendix, demonstrates that MCTS can effectively generate local trajectory improvement.

**Theorem 2.** *For a state $s \in \mathcal{S}$, let $\epsilon \leq (1 - \pi(a^*|s)) \min_{a \neq a^*}(Q_\pi^e(s, a^*) - Q_\pi^e(s, a))$. After performing $n$ MCTS searches according to the modified MCTS algorithm described in Section 4.2, we use $\pi_n(s)$ to denote the action distribution at state $s$, which is proportional to $n(s, a)$. If $|V_\pi^e(s) - V_\pi(s)| \leq \frac{\epsilon}{2\gamma}$, as $n \to \infty$, $\mathbb{E}_{a \sim \pi_n(s)}[Q_\pi(s, a)] \geq \mathbb{E}_{a \sim \pi(s)}[Q_\pi(s, a)]$, that is, $\pi_n(s)$ provides a one-step improvement over $\pi$.*

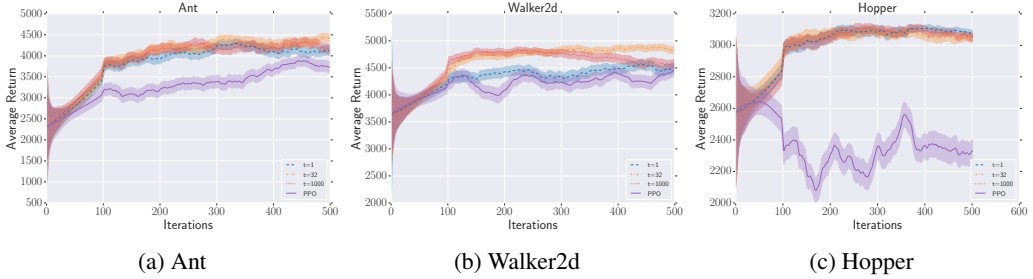

(a) Ant          (b) Walker2d          (c) Hopper

Figure 2: Average returns for continuous control tasks with different $t$ values and the PPO baseline. Experiment results are across 5 random seeded runs. Shaded area indicates $\pm 1$ standard deviation.

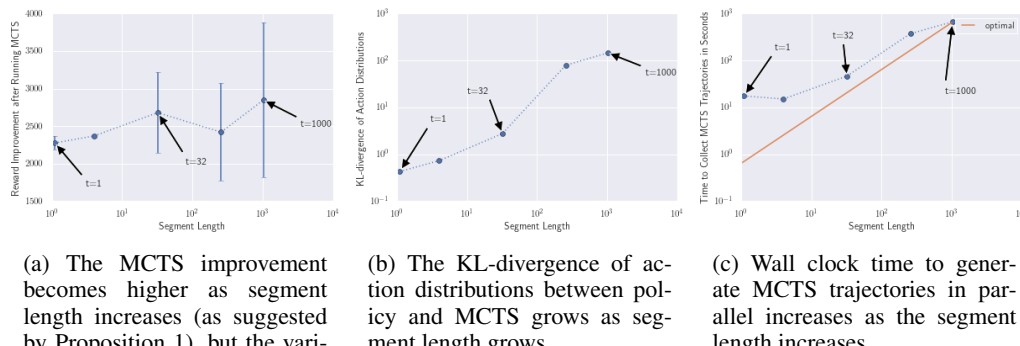

(a) The MCTS improvement becomes higher as segment length increases (as suggested by Proposition 1), but the variance also grows.

(b) The KL-divergence of action distributions between policy and MCTS grows as segment length grows.

(c) Wall clock time to generate MCTS trajectories in parallel increases as the segment length increases.

Figure 3: Tradeoff between $t$ and improved rewards from MCTS, KL-divergence, time to generate data from MCTS in the Ant environment.

# 6 EXPERIMENT

The goals of our experimental evaluation are threefold: (1) to verify that there exists a balance point for optimized length for local trajectory improvement, (2) to show that there is a trade-off between reward improvement by MCTS, and the divergence of action distributions between the policy and the expert (MCTS), (3) to demonstrate that the POLISH algorithm, by design, enables a significant speedup with a parallel implementation that is simple and efficient.

## 6.1 EXPERIMENT SETUP

We evaluate our algorithm on three environments (Ant, Walker2d, and Hopper) from OpenAI Gym (Brockman et al., 2016) with the MuJoCo physical engine (Todorov et al., 2012). For each environment, we use the standard maximum trajectory length of 1000. For each trajectory, there are multiple MCTS rollouts, each of which is distributed on a separate machine. This implementation takes advantage of the parallelizable nature of Lines 11, 12 in Algorithm 1. For instance, with a trajectory length of 1000 and a segment length of $t = 32$, we can achieve maximum parallelization by using 32 machines. Because the modified MCTS described in Section 4.2 relies on a policy during search, we need a reasonably performant policy in order for MCTS to be able to improve trajectories locally as shown in Theorem 2. Thus, we pre-train a policy with 1000 iterations of PPO (Schulman et al., 2017). In the policy optimization step (Line 15 in Algorithm 1), we use an augmented version of $L(D, \pi) = D_{KL}(\pi \| \pi^*) + H(\pi) + L_V(D, \pi)$. The first two terms, as shown in Proposition 3, quantify the disagreement between $\pi$ and $\pi^*$, which we want to minimize. Note that in practice, we do not use the 0-1 loss presented in Section 1, since fitting a distribution target leads to more stable learning as noted in (Anthony et al., 2017). The third term is a value loss similar to PPO (Schulman et al., 2017), as we need to refine the value network for better MCTS performance.

## 6.2 MAIN RESULTS

Figure 2 compares POLISH with different segment lengths: $t = 1000$ (behavioral cloning), $t = 32$ and DAgger (Ross et al., 2011), which can be regarded as $t = 1$ with dataset aggregation across iterations, as well as the vanilla PPO. Overall, MCTS-driven imitation learning achieves a higher return compared to PPO in all three environments, and by a significant margin with the best choice of segment length in particular. Regarding the choice of segment lengths, we show that our choice of $t = 32$ achieves a higher return than the two other extremes ($t = 1$ and $t = 1000$), in Ant and Walker2d which are complex environments. This provides empirical support for Equation 3, which states that there exists a sweet spot for segment length in MCTS rollouts. However, in the simple Hopper environment, it is inconclusive which segment length leads to the best return.

## 6.3 EMPIRICAL EVIDENCE OF THEORY BEHIND POLISH

To further understand the impact of segment length on the return, we show the reward improvement after running MCTS with the current policy in Figure 3a, and the KL-divergence of the action distributions between the MCTS and the policy in Figure 3b. 3a shows how the first term in Equation 3 changes as a function of segment length, and 3b depicts an approximation of the second term. For the Ant environment, we show both metrics for a select range of segment lengths of (1, 4, 32, 256, 1000). As shown in our theoretical analysis, the MCTS return improvement (over the policy) generally increases with segment length, but so does the KL-divergence. Thus, there is a sweet spot for the segment length, where both the MCTS return improvement is high and the policy can effectively learn to imitate the MCTS expert.

## 6.4 PARALLEL IMPLEMENTATION SPEEDUP

Lastly, we show that we are able to achieve a significant speedup with a parallel implementation. Our algorithm, by design, is easy to parallelize over distributed compute (Lines 11, 12 in Algorithm 1). Every MCTS rollout from an initial state is independent, and thus can be conducted on separate workers. The overhead of a parallel implementation, which arises from model synchronization across different workers and data transmission over the network, is likely insignificant compared to the workload of a MCTS rollout, especially for complex environments. Figure 3c shows the actual runtime of the parallel version of MCTS rollouts over a trajectory, compared to the ideal lower-bound which is the runtime for one single MCTS rollout at a segment length $t$. Even with a small segment length, e.g., $t = 32$, which implies a high number of parallel tasks (=32) over a fixed trajectory length (=1000), the actual runtime is close to ideal. This suggests that we can easily achieve O(10)X speedup for OpenAI Gym-like environments. The overhead outweighs the benefit only when segment length is very small.

## 7 CONCLUSION AND FUTURE WORK

In this work, we propose POLISH, a general imitation learning algorithm that provides flexible local trajectory improvement based on MCTS. We provide theoretical analysis that sheds light on the benefit of local trajectory improvement as well as a specific case study on MCTS. We validate the proposed approach by demonstrating strong empirical performance on a suite of high-dimensional continuous control problems, providing empirical support for our theory.

Model based methods have been proposed as a way to improve the sample efficiency of policy optimization (Watter et al., 2015; Finn & Levine, 2016; Hafner et al., 2018; Piergiovanni et al., 2018; Clavera et al., 2018; Sun et al., 2018b; Janner et al., 2019). However, model based policy optimization methods are vulnerable to model inaccuracy, which can lead to compounding errors for long horizon planning (Cheng et al., 2018). Nevertheless, the model can still be accurate enough for planning shorter trajectories, meaning that we can leverage it for the proposed POLISH algorithm, as it relies only on local trajectory improvement. In particular, we can learn a model of the environment and use that model to perform MCTS. As long as the learned model is reasonably accurate on short time horizons (Sun et al., 2018b), we can expect that it will be able to improve the policy.

In this work, we focused on an expert feedback collection method with a fixed local search horizon. One future direction would be to design a more adaptive approach where the length of local search horizons are learned, striking a balance between feedback quality and state distribution shift.

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

## A  PROOFS FROM SECTION 5

Proof of Proposition 1

*Proof.* Let $t' = kt$ for some positive integer $k$. Since Algorithm 1 collects initial state every $t$ and $t'$ steps, respectively. The set of starting states collected with the interval $t'$ is a subset of those collected with an interval of $t$. We will show that for the first segment of length $t'$ starting from some state $s$, $J_\pi^{0:t'}(\pi^*) \geq \sum_{i=1}^{k} \gamma^{t(i-1)} J_\pi^{t_i:t_{i+1}}(\pi^*)$. Similar proof holds for the other segments as well.

$$J_\pi^{0:t'}(\pi^*) = \sum_{i=1}^{k} \gamma^{t(i-1)} J_{\pi^*}^{t_i:t_{i+1}}(\pi^*) \tag{4}$$

$$\geq \sum_{i=1}^{k} \gamma^{t(i-1)} J_\pi^{t_i:t_{i+1}}(\pi^*) \tag{5}$$

The inequality holds term-by-term because by Assumption 1, following $\pi$ instead of $\pi^*$ generates state distributions that induce a lower valued states.

To show that the sum is maximized at $t = T$, consider any $t < T$.

$$J_\pi^{0:T}(\pi^*) = \sum_{i=1}^{T/t} \gamma^{t(i-1)} J_{\pi^*}^{t_i:t_{i+1}}(\pi^*) \tag{6}$$

$$\geq \sum_{i=1}^{T/t} \gamma^{t(i-1)} J_\pi^{t_i:t_{i+1}}(\pi^*) \tag{7}$$

It follows that the sum is maximized at $t = T$. $\qquad\square$

Proof of Proposition 2

*Proof.* Let $f(t) = \frac{t^2}{1-\gamma^t}$. To prove this proposition, it suffices to show that $f(t)$ is an increasing function for $t > 0$. First we take the derivative and get $f'(t) = \frac{2t(1-\gamma^t)+t^2\gamma^t \ln\gamma}{(1-\gamma^t)^2}$.

Let $g(t) = 2t(1 - \gamma^t) + t^2\gamma^t \ln\gamma$ be the numerator term. It is sufficient to show that $g(t) > 0$ for $t > 0$. Taking the derivative for $g(t)$, we obtain

$$\begin{aligned}
g'(t) &= 2 - 2\gamma^t - 2t\gamma^t \ln\gamma + \ln\gamma(2t\gamma^t + t^2\gamma^t \ln\gamma) \\
&= 2 - 2\gamma^t + (\ln\gamma)^2 t^2\gamma^t \\
&> 0
\end{aligned}$$

the last inequality follows from $\gamma \in (0, 1)$. So $g(t)$ is increasing, i.e., $g(t) > g(0) = 0$. It follows that $f'(t) > 0$ and $f(t)$ is an increasing function of $t$. $\qquad\square$

Proof of Proposition 3:

*Proof.*

$$\epsilon = \mathbb{E}_{s \sim d_\pi^*}[e_\pi(s)] \tag{8}$$

$$= \mathbb{E}_{s \sim d_{\pi^*}}\left[\sum_{a \in \mathcal{A}} \pi(a|s)(1 - \pi^*(a|s))\right] \tag{9}$$

$$\leq \mathbb{E}_{s \sim d_{\pi^*}}\left[\pi(a|s) \log \frac{1}{\pi^*(a|s)}\right] \tag{10}$$

$$= \mathbb{E}_{s \sim d_{\pi^*}}[D_{KL}(\pi(s)||\pi^*(s)) + H(\pi(s))] \tag{11}$$

$$\square$$

Proof of Theorem 2:

We first provide an analysis that extends theoretical guarantees of UCT (Kocsis et al., 2006; Kocsis & Szepesvári, 2006) to the setting with the modified UCT rule as in 4.2.

As usual, we connect UCT with the Upper-Confidence Bound algorithm in multi-armed bandits problem (Lai & Robbins, 1985; Agrawal, 1995; Auer et al., 2002) by framing the UCT as a non-stationary bandit problem. Specifically, we assume there are $K$ arms. At time step $t$, arm $i$ has a

reward distribution $X_{it}$ which lies in the interval $[0, 1]$ for simplicity. Unlike in traditional multi-armed bandits, we allow $X_{it}$ to vary as a function of $t$. Following the standard procedure for analyzing UCT-type algorithms in (Kocsis et al., 2006; Kocsis & Szepesvári, 2006), we assume that the reward distributions concentrate.

**Assumption 2.** *For each* $1 \leq i \leq K$, $\overline{X}_{in} = \frac{1}{n} \sum_{t=1}^{n} X_{it}$ *converges. Let* $\mu_{in} = \mathbb{E}[\overline{X}_{in}]$ *and* $\mu_i = \lim_{n \to \infty} \mu_{in}$. *Let* $\delta_{in}$ *be the bias with respect to* $\mu_i$, *i.e.,* $\delta_{in} = \mu_i - \mu_{in}$, *assume that there exists a constant* $C_p > 0$ *and an integer* $N_p$ *such that for* $n \geq N_p$, *for any* $\delta > 0$, $\Delta_n(\delta) = C_p \sqrt{n \ln(1/\delta)}$, *the following two tail probability inequalities are satisfied*

$$\mathbb{P}(\overline{X}_{is} \geq \mu_i + \Delta_n(\delta)) \leq \delta \tag{12}$$

$$\mathbb{P}(\overline{X}_{is} \leq \mu_i - \Delta_n(\delta)) \leq \delta \tag{13}$$

The UCB algorithm balances the exploration-exploitation problem by selecting arms with the highest upper confidence bound. We consider a related variant of UCB that weights the exploration term by some prior probabilities $p_i$ for arm $i$. A central quantity analyzed in multi-armed bandit problems is the regret which connects to the number of times some sub-optimal arm is chosen. Let $T_i(n)$ be a random variable representing the total number of times that arm $i$ is chosen in the first $n$ rounds. We assume a unique optimal arm exists and its related quantities will be indexed by a $*$, i.e., $p_*, \mu^*, T_*(n), \overline{X}_t^*$. We first upper bound the expected number of choices of a sub-optimal arm in terms of its sub-optimality $\Delta_i = \mu^* - \mu_i$ and the prior probability.

Since $\mu_{in}$ converges to $\mu_i$ as $n \to \infty$, $\delta_{in}$ converges to zero, so for any $\epsilon > 0$, there exists an index $N_0(\epsilon)$ such that if $t \geq N_0(\epsilon)$, then $|\delta_{it}| \leq \frac{\epsilon \Delta_i}{2}$ for every arm $i$.

**Theorem 3.** *Under Assumption 2, if we use* $c_{t,s} = 2C_p \sqrt{\frac{\ln t}{s}}$ *for the modified UCB algorithm. Fix* $\epsilon > 0$, $\mathbb{E}[T_i(n)] \leq \lceil \frac{16 p_i^2 \ln n}{(1-\epsilon)^2 \Delta_i^2} \rceil + N_0(\epsilon) + N_p + \frac{\pi^2}{6}(e^{p_i^2} + e^{p_*^2})$.

*Proof.* We follow the proof in (Kocsis et al., 2006). Define $A_0(n, \epsilon) = \min\{s | c_{t,s} \leq \frac{(1-\epsilon)\Delta_i}{2p_i}\}$. Since $c_{t,s} = 2C_p \sqrt{\frac{\ln t}{s}}$, $A_0(n, \epsilon) = \lceil \frac{16 C_p^2 p_i^2 \ln n}{(1-\epsilon)^2 \Delta_i^2} \rceil$. Let $A(n, \epsilon) = \max(A_0(n, \epsilon), N_0(\epsilon), N_p)$. Then

$$T_i(n) = 1 + \sum_{t=K+1}^{n} I(I_t = i) \tag{14}$$

$$\leq A(n, \epsilon) + \sum_{t=K+1}^{n} I(I_t = i, T_i(t-1) \geq A(n, \epsilon)) \tag{15}$$

$$\leq A(n, \epsilon) + \sum_{t=1}^{n} \sum_{s=1}^{t-1} \sum_{s'=A(n,\epsilon)}^{t-1} I(\overline{X}_s^* + p_* c_{t,s} \leq \overline{X}_{i,s'} + p_i c_{t,s'}) \tag{16}$$

We observe that $\overline{X}_s^* + p_* c_{t,s} \leq \overline{X}_{i,s'} + p_i c_{t,s'}$ implies that at least one of the following is true

$$\overline{X}_s^* + p_* c_{t,s} \leq \mu^* \tag{17}$$

$$\overline{X}_{i,s'} \geq \mu_i + p_i c_{t,s'} \tag{18}$$

$$\mu^* < \mu_i + 2 c_{t,s'} p_i \tag{19}$$

Since $s' \geq A(n, \epsilon)$ and by the construction of $A(n, \epsilon)$, $A(n, \epsilon) \geq A_0(n, \epsilon)$ and $A(n, \epsilon) \geq N_0(\epsilon)$, so the inequality (19) does not hold. Thus, either (17) or (18) must hold. So $I(\overline{X}_s^* + p_* c_{t,s} \leq \overline{X}_{i,s'} + p_i c_{t,s'}) \leq I(\overline{X}_s^* + p_* c_{t,s} \leq \mu^*) + I(\overline{X}_{i,s'} \geq \mu_i + p_i c_{t,s'})$. Now we can utilize inequalities (4) and (5) to derive upper bound on the two probabilities, so

$$I(\overline{X}_s^* + p^* c_{t,s} \leq \mu^*) = \mathbb{P}(\overline{X}_s^* \leq \mu^* - p^* c_{t,s}) \leq e^{p_*^2}/t^4 \tag{20}$$

$$I(\overline{X}_{i,s'} \geq \mu_i + p_i c_{t,s'}) = \mathbb{P}(\overline{X}_{i,s'} \geq \mu_i + p_i c_{t,s'}) \leq e^{p_i^2}/t^4 \tag{21}$$

Plugging those quantities into (8) and use the relaxation that $A(n, \epsilon) \leq A_0(n, \epsilon) + N_0(\epsilon) + N_p$, we obtain the desired result.

$\square$

It follows that the asymptotic probability of choosing a sub-optimal arm goes to zero, the same guarantee in the original UCT algorithm, the rate of convergence is multiplied by the the square of the prior probability $p_i^2$. This means that if the prior probability is informative that it assigns small probabilities to sub-optimal arms, we can identify the optimal arm faster.

Next, we consider the more drastic change to the MCTS procedure by replacing the Monte Carlo rollout step with an estimation, usually from a value network learned with a policy (Silver et al., 2017; Anthony et al., 2017). An accurate value estimation can effectively reduce the amount of lookahead necessary to identify an optimal action (Ng, 2003; Sun et al., 2018a). On the other hand, an inaccurate value estimation can mislead the MCTS process. We provide a sufficient condition under which MCTS is guaranteed to improve the policy. It follows that, compared with the true Q-function $Q_\pi(s, a)$.

$$
\begin{aligned}
|Q_\pi^e(s, a) - Q_\pi(s, a)| &= |\mathbb{E}[r(s, a) + \gamma V_\pi^e(s')] - \mathbb{E}[r(s, a) + \gamma V_\pi^e(s')]| \\
&= \gamma |\mathbb{E}[] V_\pi^e(s') - V_\pi(s')]| \\
&\leq \gamma \cdot \frac{\epsilon}{2\gamma} = \frac{\epsilon}{2}
\end{aligned}
\tag{22}
$$

Before we finally prove the main theorem, we need the following lemma stating improvement when computed with respect to the estimated $Q$-function.

**Lemma 1.** *Let $a^* = \arg\max_a Q_\pi^e(s, a)$ for a state $s$ and $\epsilon \leq (1 - \pi(a^*|s)) \min_{a \neq a^*} Q_\pi^e(s, a^*) - Q_\pi^e(s, a)$. Then as $n \to \infty$, $\mathbb{E}_{a \sim \pi_n(s)}[Q_\pi^e(s, a)] \geq \mathbb{E}_{a \sim \pi(s)}[Q_\pi^e(s, a)] + \epsilon$.*

*Proof.* As $n \to \infty$, $\pi_n(s)$ becomes a delta distribution with probability mass 1 on the $a^*$, so

$$
\mathbb{E}_{a \sim \pi_n(s)}[Q_\pi^e(s, a)] - \mathbb{E}_{a \sim \pi(s)}[Q_\pi^e(s, a)] = Q_\pi^e(s, a^*) - \mathbb{E}_{a \sim \pi(s)}[Q_\pi^e(s, a)]
\tag{23}
$$

$$
= (1 - \pi(a^*|s))Q_\pi^e(s, a^*) - \mathbb{E}_{a \sim \pi(s), a \neq a^*}[Q_\pi^e(s, a)]
\tag{24}
$$

$$
\geq (1 - \pi(a^*|s)) \min_{a \neq a^*} Q_\pi^e(s, a^*) - Q_\pi^e(s, a)
\tag{25}
$$

$$
\geq \epsilon
\tag{26}
$$

$\square$

Finally, we prove the main theorem,

$$
\mathbb{E}_{a \sim \pi_n(s)}[Q_\pi(s, a)] \geq \mathbb{E}_{a \sim \pi_n(s)}[Q_\pi^e(s, a) - \frac{\epsilon}{2}]
\tag{27}
$$

$$
\geq \mathbb{E}_{a \sim \pi_n(s)}[Q_\pi^e(s, a)] - \frac{\epsilon}{2}
\tag{28}
$$

$$
\geq \mathbb{E}_{a \sim \pi(s)}[Q_\pi^e(s, a)] + \epsilon - \frac{\epsilon}{2}
\tag{29}
$$

$$
= \mathbb{E}_{a \sim \pi(s)}[Q_\pi^e(s, a) + \frac{\epsilon}{2}]
\tag{30}
$$

$$
\geq \mathbb{E}_{a \sim \pi(s)}[Q_\pi(s, a)]
\tag{31}
$$

