# OpenReview forum: "Policy Optimization by Local Improvement through Search"
_ICLR.cc/2020/Conference — Reject_

### Official Review · AnonReviewer1 · 2019-10-06
**Official Blind Review #1**

**Rating:** 3

**Review:**

This paper proposes POLISH, a reinforcement learning learning algorithm based on imitating partial trajectories produced by an MCTS procedure. The intuition behind this idea is that behavioral cloning suffers from distribution shift over time, and using MCTS allows imitation learning to be done on states closer to the policy's state distribution, which the authors justify using techniques similar to DAgger. The authors evaluate this method on continuous OpenAI Gym tasks, and show that it consistently beats a PPO baseline.

Overall, my decision for this paper errs on the side of reject. This primarily comes from the fact that the writing is unclear to me, and this algorithm needs both access to an expert and a reward function, which is a setting that I'm not sure is very applicable in practice. Additionally, the experimental results seem fairly weak. In the imitation learning setting, there appears to be little difference between behavioral cloning, DAgger, and an intermediate segment length. In the reinforcement learning setting, the PPO baseline seems to be unfair, which I detail below.

The writing is confusing to me as it mixes two seemingly distinct problem settings (imitation learning and reinforcement learning) and interferes with my full understanding of the motivation of the paper. My current guess is that this paper is primarily aimed towards policy optimization in a reinforcement learning setting. The algorithm can start from scratch with a random initial policy, and optimize the policy to maximize total returns. However, much of the paper is written as if the setting were imitation learning, where expert advice is available. If this paper is primarily aimed at imitation learning paper, I am unsure of the advantage of using this method over DAgger. My hypothesis is that the primary use-case of POLISH over DAgger is when the provided expert is suboptimal, and using MCTS allows the policy to improve beyond the expert. However, this point is not provided in the paper.

For the experiments, I'm not sure that PPO is a directly comparable baseline.
1) Were queries to the simulator used during MCTS accounted for when measuring sample complexity? (Figure 2). Being able to query the environment without cost is a significant advantage to POLISH in terms of sample complexity.
2) Additionally, it was stated that POLISH had access to a pre-trained policy, which is additional information that PPO cannot exploit. A reasonable comparison could be to initialize the PPO agent from that pre-trained policy, or to not give POLISH access to the pre-trained policy.

For related work, I would argue that an important class of algorithms to mention are RL methods based on imitating some sort of policy improvement procedure. This includes work such as (not exhaustive) self-imitation learning (Oh 2019), the cross-entropy method, guided policy search (Levine 14), reward-weighted regression (Peters 07) and UREX (Nachum 17).

**Experience Assessment:**

I have published in this field for several years.

**Review Assessment: Checking Correctness Of Derivations And Theory:**

I assessed the sensibility of the derivations and theory.

**Review Assessment: Checking Correctness Of Experiments:**

I carefully checked the experiments.

**Review Assessment: Thoroughness In Paper Reading:**

I read the paper thoroughly.

---

> ### Author Response · Authors · 2019-11-15
> **Response to Review #1**
>
> We thank you for your review and suggestions for improvement.
>
> Regarding your concern on “both access to an expert and a reward function”:
> We want to emphasize that the only prerequisite we need to apply POLISH is access to the environment. The expert, in our case, is not a predefined policy. Rather, it is built via MCTS dynamically. This is applicable in cases where a simulator is available, for example games and robotics.
>
> Regarding “mixes two seemingly distinct problem settings (imitation learning and reinforcement learning)”:
> We do not require a predefined expert policy, rather we build our own expert through interactions with the environment by performing MCTS. It is a standard approach in the absence of an expert [4, 5, 6].
>
> We wish to point out that combining both learning settings is a popular approach to policy optimization [1, 2, 3, 4]. In this paper, we provide a novel approach of combining IL and RL through the application of local policy improvement. Given a current policy, we showed that using MCTS to plan with the ***current policy*** can result in a better policy. This better policy plays the role of an “expert”. Your understanding is correct in the primary use-case of POLISH is when the current policy is sub-optimal, which is why we are performing policy optimization in the first case.
>
> Regarding “not sure that PPO is a directly comparable baseline”:
> We agree and we could improve the presentation of the experiment section. The main comparison we are interested in is to compare different values of t (the rollout horizon for MCTS). PPO is a reference for a pure reinforcement learning algorithm to show that by imitating an MCTS policy, we can improve over pure reinforcement learning approach.
>
> We apologize for the confusion. Both POLISH and PPO are initialized from the same pre-trained policy.

---

### Official Review · AnonReviewer2 · 2019-10-17
**Official Blind Review #2**

**Rating:** 1

**Review:**

The paper was confusing and difficult to read.  I think it is trying to devise a methodology that improves upon a given expert policy, but I am not confident whether if this is the objective.  The main contribution of the paper is the algorithm POLISH on page 5.  However, it was unclear how MCTS with UCT is used with the expert policy in the algorithm.  The first mention of the 0-1 loss objective is in section 4.1, and Algorithm 1 on page 5 claims to minimize this loss on line 15.  However, in the experiment, the loss function is then switched to another L(D, \pi) = D_{KL}(\pi || \pi*) + H(\pi) + Lv(D, \pi).  Why the switch?  And what is the definition of the third term?  By examining the experimental results, Algorithm 1 seems to outperform PPO.  Since the expert policy is known and given to Algorithm 1, I speculate Algorithm 1 is only replicating the expert policy, which would have outperformed PPO that has to learn from scratch.  Thus, the comparison does not seem fair.  It would be interesting to see how POLISH compares with the expert policy.

Other comments:

* Page 1 in Introduction: “However, these models suffer from the problem that even small difference between the learned policy and the expert behavior can lead to a snow-balling effect, where the state distribution diverges to a place where the behaviour of the policy is now meaningless since it was not trained that part of space”.  Do you mean that the algorithm diverges instead of the state distribution diverges?   The agent may incur errors in a space that has not been observed, but it should be able to learn eventually.  So, what is causing divergence in states that have not yet encountered?

* Page 4 in The POLISH Algorithm Main Algorithm: the 0-1 loss function is defined to be “L(D, \pi) = 1/|D| \sum_{s,a*}\in D (I(\pi(s) \neq a*)), where a* is the expert policy’s selected action.  I think it makes more sense to write “L(D, \pi)" as 1/|D| \sum_{s,a*}\in D I ( a \neq a* : a ~ \pi(s)).  Also, in RL, we don't say that the policy receives a reward, but rather the agent receives a reward.

**Experience Assessment:**

I do not know much about this area.

**Review Assessment: Checking Correctness Of Derivations And Theory:**

I assessed the sensibility of the derivations and theory.

**Review Assessment: Checking Correctness Of Experiments:**

I assessed the sensibility of the experiments.

**Review Assessment: Thoroughness In Paper Reading:**

I read the paper at least twice and used my best judgement in assessing the paper.

---

> ### Author Response · Authors · 2019-11-15
> **Response to Review #2**
>
> We thank you for taking the time to review our paper. We will improve our writing to make it more accessible to the general audience. We answer a few clarification questions here.
>
> The main research question is: given an expert policy (e.g., MCTS), how can we best collect demonstrations from expert rollouts for most efficient imitation learning? Our main contribution is the POLISH algorithm where we collect local trajectory improvements via rolling out the expert policy for short time horizons.
>
> Regarding “how MCTS with UCT is used”:
> We use MCTS with the UCT rule [1] to generate local improvements of an existing trajectory. This is similar to how AlphaZero [2] works.
>
> Regarding the 0-1 loss and the empirical loss used in the experiment:
> The reason we switch is mentioned in the paragraph above Proposition 2. The loss in our experiment basically matches a distribution instead of a single action as is the case for the 0-1 loss. The third term is a value loss term defined in the last sentence of section 6.1. This term is similar to the value loss term in the PPO algorithm [3].
>
> Regarding state distribution divergence:
> This is a common problem with imitation learning [4]. Since the expert demonstration data are collected from the state distribution induced from the expert policy, once the learned policy deviates from this trained state distribution, it is essentially encountering states not seen in the training. While it is possible with enough demonstrations, we can minimize the amount of unseen states. It is practically impossible to collect that many demonstrations. Thus, this is a central challenge in imitation learning.
>
> [1]: Browne, Cameron B., et al. "A survey of monte carlo tree search methods." IEEE Transactions on Computational Intelligence and AI in games 4.1 (2012): 1-43.
> [2]: Silver, David, et al. "Mastering chess and shogi by self-play with a general reinforcement learning algorithm." arXiv preprint arXiv:1712.01815 (2017).
> [3]: Schulman, John, et al. "Proximal policy optimization algorithms." arXiv preprint arXiv:1707.06347 (2017).
> [4]: Ross, Stéphane, Geoffrey Gordon, and Drew Bagnell. "A reduction of imitation learning and structured prediction to no-regret online learning." Proceedings of the fourteenth international conference on artificial intelligence and statistics. 2011.

---

### Official Review · AnonReviewer3 · 2019-10-23
**Official Blind Review #3**

**Rating:** 3

**Review:**


[Summary]
This paper proposes POLISH, an imitation learning algorithm that provides a balance between Behavioral Cloning (BC) and DAgger. The algorithm reduces the mismatch between the target policy and an expert policy on states obtained from starting at the target policy's state distribution and following the expert policy for a time segment of t steps. The claim is that a suitable t will keep the training states close to the target policy's state distribution and avoid the compounding errors that arise when the agent drifts away from its training distribution. The paper also explores the possibility of policy optimization by replacing the pre-defined expert policy in POLISH with a policy derived from Monte Carlo Tree Search. Theoretical and empirical analyses in the paper studies the effect of t and MCTS planning in POLISH on policy improvement.

[Decision]
A clear study of the dimension between BC and DAgger is a useful contribution to the literature and an algorithm that effectively solves the distributional shift problem in BC will be of high practical value. However, the results in this paper do not support the claim that a reasonable time segment length in POLISH alleviates this problem. The theory shows a bound on the performance of the target policy that varies with t, but it is not clear if a suitable t is better than the two extremes, i.e., BC and DAgger. The experiments section is limited and, on two out of the three tasks, there is no considerable difference between the performance of POLISH, BC, and DAgger. I am leaning towards rejecting this paper.

[Explanation]
In Section 5, Theorem 1 shows the effect of t, the length of time segments, on the performance and the target policy by providing a bound. If this theorem is motivating a middle ground between BC and DAgger, it needs to show that a value of t other than the extremes will maximize the bound. The bound in the paper consists of a positive term and a negative term, both of which grow with t. It is then concluded that a balance point will maximize the overall bound. I do not see how it follows that this balance point is a middle ground and not an extreme value. If, for example, the negative term grows faster than the positive term, then DAgger (t=1) will be have the best performance according to this theorem.

It is not clear how the bound in Theorem 1 is comparing the performance of algorithms with different values of t. For a fixed policy, one can obtain different bounds by choosing different values of t. These bounds will have different values of \epsilon. The state distribution for \epsilon is the distribution of states visited in a limited time segment (which depends on the target policy, the expert policy, and the segment length). Using \epsilon (or \epsilon_i) in the bound drops the relationship between \epsilon and the length of time segments, and hides the fact that different algorithms are minimizing different errors.

The equality in Eq 2 is not obvious to me. J(\pi) is and expectation under the discounted visit distribution. For example, if gamma is small, then the states in the start state distribution will have a higher weight in J(\pi) while the right-hand side sums over the expected performance in all time segments equally. I believe the later terms in the sum should also be discounted.

Does changing t also affect the MCTS step and the expert policy obtained from it? If so, it is possible that a suitable t will result in better performance because the MCTS step finds a stronger expert policy, and not because the imitation learning step better reduces the error.

Does the MCTS step use the perfect simulator or a learned model in the experiments? If POLISH, unlike PPO, has access to the MDP, the comparison of these two methods is not fair.

In Fig 2 (a) and (c), the curves for t=1, t=32, and t=1000 overlap through most of the training process. This is not conclusive evidence that a sweet spot for t results in better performance. An experiment on a simpler setting with more runs may elucidate the effect of t on the performance.

In 6.3, what does the reward improvement after running MCTS with the current policy precisely mean, and how does this correspond to the first term in the bound, i.e., the sum of the expected performance of \pi_* over time segments?


[Minor comments]
- I suggest adding the process of obtaining an expert policy through MCTS to Algorithm 1. It is hard to understand the process without a clear step-by-step description.
- How is t=32 chosen for the experiments in Fig 2?
-------------------
After rebuttal: I have read the authors' response and the other reviews. The rebuttal addresses my questions and concerns about clarity of presentation. However, I am still not convinced by the evidence in this paper. On two out of the three environments, the performance of the three values of t is not much different through the training and this is not because of the advantage over PPO. On Ant, for example, the curves for t=32 and t=1000 are not even one std apart through most of the training.


**Experience Assessment:**

I have read many papers in this area.

**Review Assessment: Checking Correctness Of Derivations And Theory:**

I assessed the sensibility of the derivations and theory.

**Review Assessment: Checking Correctness Of Experiments:**

I carefully checked the experiments.

**Review Assessment: Thoroughness In Paper Reading:**

I read the paper at least twice and used my best judgement in assessing the paper.

---

> ### Author Response · Authors · 2019-11-15
> **Response to Review #3**
>
> Thank you for your detailed review and your suggestion for improvements. Thank you for recognizing the contribution of our submission.
>
> Regarding “a value of t other than the extremes will maximize the bound”:
> Firstly, Theorem 1 does not concern the effect of t, it is an existing result we used to derive the consideration for t in equation (3).
>
> It certainly is desirable to have an analytic form for what value of t maximizes the bound. However, without making additional assumptions on how J(\pi^*) grows, it is not possible to derive an expression for an optimal t. Thus we phrase it as “there can be a balance point” and treat it as the motivation for experimenting with different t’s in our empirical studies.
>
> Regarding “these bounds will have different values of \epsilon”:
> Your understanding is correct. We assumed the \epsilon’s remained constant across different values of t to focus on the main point, the relationship of the bound with respect to different t. Assuming constant \epsilon is an approximation to what happens in practice.
>
> In practice, we found that it became harder to imitate long time horizon MCTS rollouts. If we refer back to Proposition 2, the KL-divergence plus the entropy is an upper bound on the \epsilon. In Figure 3 (b), we showed that the KL-divergence term grows as t grows and in practice the entropy term remains comparable across different values of t. Thus the upper bound grows as t grows.
>
> Regarding discounting in Equation (2):
> Thanks for catching this! We have updated the analysis with the correct discountings.
>
> Regarding “does changing t also affect the MCTS step and the expert policy obtained”:
> Changing t does not change the MCTS step. It changes how long we obtain the MCTS demonstrations for while maintaining the same MCTS policy.
>
> Regarding “does the MCTS step use the perfect simulator”:
> We use the environment interactions during MCTS. The main contribution is to find a sweet point for the value of t when MCTS is used in the policy training. For the comparison with different values of t, the number of environment interactions is the same so the comparison is fine across values of t.
>
> The comparison with PPO is to provide a pure reinforcement learning baseline and shows the advantage of the proposed approach.
>
> Regarding curves are overlapping:
> We wish to point out that in the Ant and Walker that the advantage of t=32 over the next best value of t on the mean return is around 300, which is significant. The curves look largely overlapping is because of the advantage, when compared with the advantage over PPO, looks small. We will improve the presentation in the figures to make the differences more obvious.
>
> Regarding “in 6.3, what does the reward improvement … mean”:
> It is the first term in the bound minus J(\pi) where \pi is the current policy. Figure 6.3 shows how much better the MCTS expert is over the current policy. Since the MCTS policy does not change when t varies, this shows the effect of changing initial state distributions has on the first term in the bound.
>
> Regarding minor comments:
> We will add more details on obtaining an expert policy through MCTS.
> t = 32 is chosen because it is approximately halfway between 1 and 1000 in terms of multiplicative factors.

---

### Decision · Program_Chairs · 2019-12-19

**Decision:**

Reject

**Comment:**

Thanks for your detailed responses to the reviewers, which helped us a lot to better understand your paper.
However, given that the current manuscript still contains many unclear parts, we decided not to accept the paper. We hope that the reviewers' comments help you improve your paper for potential future submission.